# EXCLUSIVE SUPERMASK SUBNETWORK TRAINING FOR CONTINUAL LEARNING

## ABSTRACT

Continual Learning (CL) methods mainly focus on avoiding catastrophic forgetting and learning representations that are transferable to new tasks. Recently, Wortsman et al. (2020) proposed a CL method, SupSup, which uses a randomly initialized, fixed base network (model) and finds a *supermask* for each new task that selectively keeps or removes each weight to produce a *subnetwork*. They prevent forgetting as the network weights are not being updated. Although there is no forgetting, the performance of supermask is sub-optimal because fixed weights restrict its representational power. Furthermore, there is no accumulation or transfer of knowledge inside the model when new tasks are learned. Hence, we propose ExSSNeT (Exclusive Supermask SubNEtwork Training), that performs *exclusive* and *non-overlapping* subnetwork weight training. This avoids conflicting updates to the shared weights by subsequent tasks to improve performance while still preventing forgetting. Furthermore, we propose a novel KNN-based Knowledge Transfer (KKT) module that dynamically initializes a new task's mask based on previous tasks for improving knowledge transfer. We demonstrate that ExSSNeT outperforms SupSup and other strong previous methods on both text classification and vision tasks while preventing forgetting. Moreover, ExSSNeT is particularly advantageous for sparse masks that activate 2-10% of the model parameters, resulting in an average improvement of 8.3% over SupSup. Additionally, ExSSNeT scales to a large number of tasks (100) and our KKT module helps to learn new tasks faster while improving the overall performance.[1]

## 1 INTRODUCTION

In artificial intelligence, the overarching goal is to develop autonomous agents that can learn to accomplish a set of tasks. Continual Learning (CL) (Ring, 1998; Thrun, 1998; Kirkpatrick et al., 2017) is a key ingredient for developing agents that can accumulate expertise on new tasks. However, when a model is sequentially trained on tasks $t_1$ and $t_2$ with different data distributions, the model's ability to extract meaningful features for the previous task $t_1$ degrades. This loss in performance on the previously learned tasks, is referred to as *catastrophic forgetting* (CF) (McCloskey & Cohen, 1989; Zhao & Schmidhuber, 1996; Thrun, 1998; Goodfellow et al., 2013). Forgetting is a consequence of two phenomena happening in conjunction: (1) not having access to the data samples from the previous tasks, and (2) multiple tasks sequentially updating shared model parameters resulting in conflicting updates, which is called as *parameter interference* (McCloskey & Cohen, 1989).

Recently, some CL methods avoid parameter interference by taking inspiration from the *Lottery Ticket Hypothesis* (Frankle & Carbin, 2018) and *Supermasks* (Zhou et al., 2019) to exploit the expressive power of sparse subnetworks. Zhou et al. (2019) observed that the number of sparse subnetwork combinations is large enough (combinatorial) that even within randomly weighted neural networks, there exist *supermasks* that achieve good performance. A supermask is a sparse binary mask that selectively keeps or removes each connection in a fixed and randomly initialized network to produce a subnetwork with good performance on a given task. We call this the subnetwork as *supermask subnetwork* that is shown in Figure 1, highlighted in red weights. Building upon this idea, Wortsman et al. (2020) proposed a CL method, *SupSup*, which initializes a network with fixed and random weights

---

[1]Our code is uploaded as supplementary material.

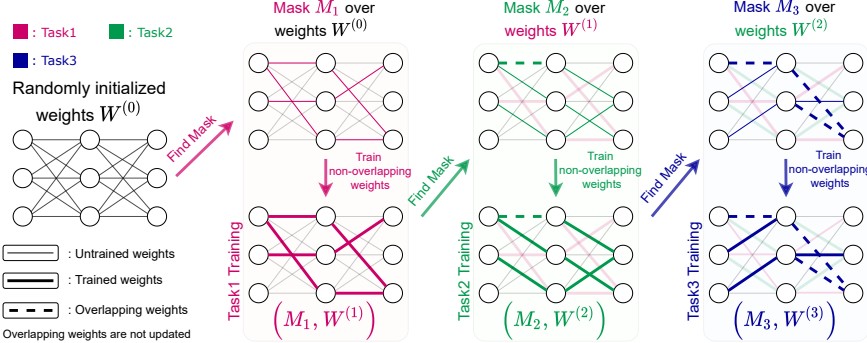

**Figure 1:** EXSSNET diagram. We start with random weights $W^{(0)}$. For task 1, we first learn a supermask $M_1$ (the corresponding subnetwork is marked by red color, column 2 row 1) and then train the weight corresponding to $M_1$ resulting in weights $W^{(1)}$ (bold red lines, column 1 row 2). Similarly, for task 2, we learn the mask $M_2$ over fixed weights $W^{(1)}$. If mask $M_2$ weights overlap with $M_1$ (marked by bold dashed green lines in column 3 row 1), then only the non-overlapping weights (solid green lines) of the task 2 subnetwork are updated (as shown by bold and solid green lines column 3 row 2). These already trained weights (bold lines) are not updated by any subsequent task. Finally, for task 3, we learn the mask $M_3$ (blue lines) and update the solid blue weights.

and then learns a different supermask for each new task. This allows them to prevent catastrophic forgetting (CF) as there is no parameter interference (because the model weights are fixed).

Although SupSup (Wortsman et al., 2020) prevents CF, there are some problems to using supermasks for CL: (1) Fixed random model weights in SupSup limits the supermask subnetwork's representational power resulting in sub-optimal performance. As shown in Figure 2, for a single task the test accuracy of SupSup is approximately 10% worse compared to a *fully trained model* where all model weights are updated. As a possible remedy, one could try to naively train the weights corresponding to supermask subnetworks of different tasks; however, it can lead to CF as shown in Figure 3. This happens because subnetworks for different tasks can overlap and training subnetworks weights might result in parameter interference. (2) When learning a task, there is no mechanism for transferring knowledge from previously learned tasks to better learn the current task. Moreover, the model is not accumulating knowledge over time as they weights are not being updated.

We overcome the aforementioned issues, we propose our method, EXSSNET (Exclusive Supermask SubNEtwork Training), pronounced as *'excess-net'*, which first learns a mask for a task and then selectively trains a subset of weights from the supermask subnetwork. We train the weights of this subnetwork via *exclusion* that avoids updating parameters from the current subnetwork that have already been updated by any of the previous tasks. This helps us to prevent forgetting. This procedure is demonstrated in Figure 1 for learning three tasks sequentially. Training the supermask subnetwork's weights increases its representational power and allows EXSSNET to encode task-specific knowledge inside the subnetwork. This solves the first problem and allows EXSSNET to perform comparable to a fully trained network on individual tasks; and when learning multiple tasks, the exclusive subnetwork training improves the performance of each task while still preventing forgetting.

To address the second problem of knowledge transfer, we propose a $k$-nearest neighbors based knowledge transfer (KKT) module that transfers relevant information from the previously learned tasks to improve performance on new tasks while learning them faster. Our KKT module uses KNN classification to select a subnetwork from the previously learned tasks that has better than random predictive power for the current task and use it as a starting point to learn the new tasks.

Next, we show our method's advantage by experimenting with both natural language and vision tasks. For natural language, we evaluate on WebNLP classification tasks (de Masson d'Autume et al., 2019; Huang et al., 2021) and GLUE benchmark tasks (Wang et al., 2018), whereas, for vision, we evaluate on SplitMNIST (Zenke et al., 2017; De Lange & Tuytelaars, 2021), SplitCIFAR100 (Chaudhry et al., 2018; De Lange & Tuytelaars, 2021), and SplitTinyImageNet (Buzzega et al., 2020) datasets. We show that for both language and vision domains, EXSSNET outperforms multiple strong and recent continual learning methods based on replay, regularization, distillation, and parameter isolation. For the vision domain, EXSSNET outperforms the strongest baseline by 4.8% and 1.4% on SplitCIFAR and SplitTinyImageNet datasets respectively, while surpassing multitask model and bridging the gap to training *individual* models for each task. In addition, for GLUE datasets, EXSSNET is 2%

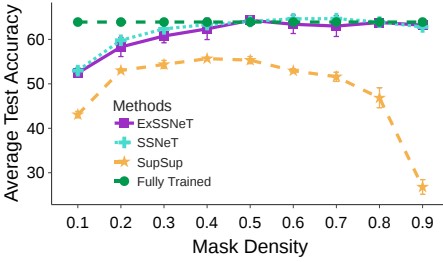 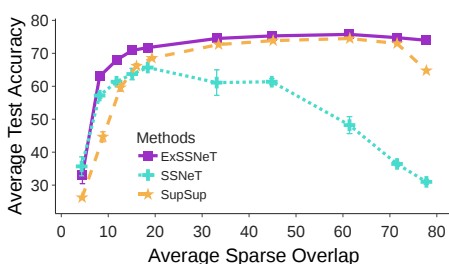

**Figure 2:** Test accuracy versus the mask density for 100-way CIFAR100 classification. Results are averaged over three seeds.

**Figure 3:** Average Test accuracy on five 20-way tasks from SplitCIFAR100 as a function of sparse overlap. Results are averaged over three seeds.

better than the strongest baseline methods and surpasses the performance of multitask learning (that has access to all the tasks at once and does not suffer from CF). Moreover, EXSSNET obtains an average improvement of 8.3% over SupSup for sparse masks with $2 - 10\%$ of the model parameters and scales to a large number of tasks (100). Furthermore, EXSSNET with the KKT module learns new tasks in as few as 30 epochs compared to 100 epochs without it, while achieving 3.2% higher accuracy on the SplitCIFAR100 dataset. In summary, our contributions are listed below:

- We propose a simple and novel method to improve mask learning by combining it with exclusive subnetwork weight training to improve CL performance while preventing CF.

- We propose a KNN-based Knowledge Transfer (KKT) module for supermask initialization that dynamically identifies previous tasks to transfer knowledge to learn new tasks better and faster.

- Extensive experiments on language and vision tasks show that EXSSNET outperforms strong baselines and is comparable to multitask model for language tasks while surpassing it for vision tasks. Moreover, EXSSNET works well for sparse masks and scales to a large number of tasks.

## 2 MOTIVATION

Using sparsity for CL is an effective technique to learn multiple tasks, i.e., by encoding them in different subnetworks inside a single model. SupSup (Wortsman et al., 2020) is an instantiation of this that initializes the network weights randomly and then learns a separate supermask for each task as shown in the SupSup diagram in Appendix Figure 7. They prevent CF because the weights of the network are fixed and never updated. However, this is a crucial problem as discussed below.

**Problem 1 - Sub-Optimal Performance of Supermask:** Although fixed network weights in SupSup prevent CF, this also restricts the network's representational capacity, leading to worse performance compared to a fully trained network. We demonstrate this in Figure 2, where we report the test accuracy of different methods with respect to the fraction of network parameters selected by the mask, i.e., the *mask density* for an underlying ResNet18 model (Wortsman et al., 2020) on a *single 100-way classification* on CIFAR100 dataset. The fully trained ResNet18 model (dashed green line) achieves an accuracy of 63.9%. Similar to Zhou et al. (2019), we observe that the performance of SupSup (yellow dashed line) is at least 8.3% worse compared to a fully trained model. As a possible *partial* remedy, we propose a simple solution, SSNET (Supermask SubNEtwork Training), that first finds a subnetwork for a task and then trains the subnetwork's weights. This increases the representational capacity of the subnetwork because there are more trainable parameters. For a single task, the test accuracy of SSNET is better than SupSup for all mask densities and matches the performance of the fully trained model beyond a density threshold. But as shown below, when learning multiple tasks sequentially, SSNET gives rise to parameter interference that results in CF.

**Problem 2 - Parameter Interference Due to Subnetwork Weight Training for Multiple Tasks:** Next, we demonstrate that when learning multiple tasks sequentially, SSNET can still lead to CF. In Figure 3, we report the average test accuracy versus the fraction of overlapping parameters between the masks of different tasks, i.e., the *sparse overlap* (see Equation 1) for five different 20-way classification tasks from SplitCIFAR100 dataset with ResNet18 model. We observe that SSNET outperforms SupSup for lower sparse overlap but as the sparse overlap increases, the performance declines because the supermask subnetworks for different tasks have more overlapping (common)

weights (bold dashed lines in Figure 1). This leads to higher parameter interference resulting in increased forgetting which suppresses the gain from subnetwork weight training.

To address this second problem, our final proposed method, EXSSNET selectively trains a subset of the weights in the supermask subnetwork to prevent parameter interference. When learning multiple tasks, this prevents CF resulting in strict improvement in performance over SupSup as shown in Figure 3 while having the representational power to bridge the gap with fully trained models, see Figure 2.

## 3 METHOD

As shown in Figure 1, when learning a new task $t_i$, EXSSNET follows three steps: (1) We learn a supermask $M_i$ for the task; (2) We use all the previous tasks' masks $M_1, \ldots, M_{i-1}$ to create a free parameter mask $M_i^{free}$, that finds the parameters selected by the mask $M_i$ that were not selected by any of the previous masks; (3) We update the weights corresponding to the mask $M_i^{free}$ as this avoids parameter interference. Now, we formally describe all the step of our method EXSSNET (Exclusive Supermask SubNEtwork Training) for a Multi-layer perceptron (MLP).

**Notation:** When finding supermasks and training subnetworks weight, we can treat each layer $l$ of an MLP network separately. An intermediate layer $l$ has $n_l$ nodes denoted by $\mathcal{V}^{(l)} = \{v_1, \ldots, v_{n_l}\}$. For a node $v$ in layer $l$, let $\mathcal{I}_v$ denote its input and $\mathcal{Z}_v = \sigma(\mathcal{I}_v)$ denote its output, where $\sigma(.)$ is the activation function. Given this notation, $\mathcal{I}_v$ can be written as $\mathcal{I}_v = \sum_{u \in \mathcal{V}^{(l-1)}} w_{uv} \mathcal{Z}_u$, where $w_{uv}$ is the network weight connecting node $u$ to node $v$. The complete network weights for the MLP are denoted by $W$. When training the task $t_i$, we have access to the supermasks from all previous tasks $\{M_j\}_{j=1}^{i-1}$ and the model weights $W^{(i-1)}$ obtained after learning task $t_{i-1}$.

### 3.1 EXSSNET: EXCLUSIVE SUPERMASK SUBNETWORK TRAINING

**Finding Supermasks:** Following Wortsman et al. (2020), we use the algorithm of Ramanujan et al. (2019) to learn a supermask $M_i$ for the current task $t_i$. The supermask $M_i$ is learned with respect to the underlying model weights $W^{(i-1)}$ and the mask selects a fraction of weights that lead to good performance on the task without training the weights. To achieve this, we learn a score $s_{uv}$ for each weight $w_{uv}$, and once trained, these scores are thresholded to obtain the mask. Here, the input to a node $v$ is $\mathcal{I}_v = \sum_{u \in \mathcal{V}^{(l-1)}} w_{uv} \mathcal{Z}_u m_{uv}$, where $m_{uv} = h(s_{uv})$ is the binary mask value and $h(.)$ is a function which outputs 1 for top-$k\%$ of the scores in the layer with $k$ being the mask density. Next, we use a straight-through gradient estimator (Bengio et al., 2013) and iterate over the current task's data samples to update the scores for the corresponding supermask $M_i$ as follows,

$$s_{uv} = s_{uv} - \alpha \hat{g}_{s_{uv}} \; ; \; \hat{g}_{s_{uv}} = \frac{\partial \mathcal{L}}{\partial \mathcal{I}_v} \frac{\partial \mathcal{I}_v}{\partial s_{uv}} = \frac{\partial \mathcal{L}}{\partial \mathcal{I}_v} w_{uv} \mathcal{Z}_u$$

**Finding Exclusive Mask Parameters:** Given a learned mask $M_i$, we use all the previous tasks' masks $M_1, \ldots, M_{i-1}$ to create a free parameter mask $M_i^{free}$, that finds the parameters selected by the mask $M_i$ that were not selected by any of the previous masks. We do this by – (1) creating a new mask $M_{1:i-1}$ containing all the parameters already updated by any of the previous tasks by taking a union of all the previous masks $\{M_j\}_{j=1}^{i-1}$ by using the logical *or* operation, and (2) Then we obtain a mask $M_i^{free}$ by taking the intersection of all the network parameters not used by any previous task which is given by the negation of the mask $M_{1:i-1}$ with the current task mask $M_i$ via a logical *and* operation. Next, we use this mask $M_i^{free}$ for the exclusive supermask subnetwork weight training.

**Exclusive Supermask Subnetwork Weight Training:** For training the subnetwork parameters for task $t_i$ given the free parameter mask $M_i^{free}$, we perform the forward pass on the model after applying the mask $M_i^{free}$ as $model(x, W \odot M_i^{free})$, where $\odot$ is the element-wise multiplication. Hence, during the weight training, only the parameters corresponding to the mask $M_i^{free}$ are updated because the gradient value is 0 for all the weights $w_{uv}$ where $m_{uv}^{free} = 0$. While during the inference on task $t_i$ we use the mask $M_i$. In contrast, SSNET uses the task mask $M_i$ both during the training and inference as $model(x, W^{(i-1)} \odot M_i)$. This updates all the parameters in the mask including the parameters that are already updated by previous tasks that result in CF. Therefore, in cases where the sparse overlap is high,

ExSSNeT is preferred over SSNeT. To summarize, ExSSNeT circumvents the CF issue of SSNeT while benefiting from the subnetwork training to improve overall performance as shown in Figure 3.

### 3.1.1 Space, Time, and Memory Complexity of ExSSNeT

For training, we store an additional set of scores on GPU with size as the model weight. The additional GPU memory required is a small fraction because the model activations account for a huge fraction of the total GPU memory. Our runtime is similar to training the weight of a model with $< 5\%$ overhead due to the logical operations on masks and masking weight during the forward passes. For training time comparisons refer to Appendix Table 10. On the disk, we need to store $k * |W|$ updated weights of 32-bits and boolean mask which takes 1-bit for each parameter. Hence, we take $max(|W| * k * t, |W|) * 32 + |W| * 1$ bits in total as in the worst case we need to store all $|W|$ model weights.

### 3.2 KKT: Knn-Based Knowledge Transfer Module

When learning multiple tasks, it is a desired property to transfer information learned by the previous tasks to achieve better performance on new tasks and to learn them faster (Biesialska et al., 2020). Hence, we propose a K-Nearest Neighbours (KNN) based knowledge transfer (KKT) module that uses KNN classification to dynamically find the most relevant previous task (Veniat et al., 2021) to initialize the supermask for the current task. To be more specific, before learning the mask $M_i$ for the current task $t_i$, we randomly sample a small fraction of data from task $t_i$ and split it into a train and test set. Next, we use the trained subnetworks of each previous task $t_1, \ldots, t_{i-1}$ to obtain features on this sampled data. Then we learn $i - 1$ independent KNN-classification models using these features. Then we evaluate these $i - 1$ models on the sampled test set to obtain accuracy scores which denote the predictive power of features from each previous task for the current task. Finally, we select the previous task with the highest accuracy on the current task. If this accuracy is better than random then we use its mask to initialize the current task's supermask. This enables ExSSNeT to transfer information from the previous task to learn new tasks better and faster.

## 4 Experiments

### 4.1 Experimental Setup and Training Details

**Datasets:** For natural language domain, we follow the shared text classification setup of IDBR (Huang et al., 2021), LAMOL (Sun et al., 2019), and MBPA++ (De Lange et al., 2019) to sequentially learn five text classification tasks; (1) Yelp Sentiment analysis (Zhang et al., 2015); (2) DBPedia for Wikipedia article classification (Mendes et al., 2012) (3) Yahoo! Answer for Q&A classification (Chang et al., 2008); (4) Amazon sentiment analysis (McAuley & Leskovec, 2013) (5) AG News for news classification (Zhang et al., 2015). We call them WebNLP classification tasks for easier reference. While comparing with the previous state-of-the-art text methods, we use the same training and test set as IDBR and LAMOL containing 115,000/500/7,600 Train/Val/Test examples. For our ablation studies, we follow IDBR and use a sampled dataset, please see Appendix Table 7 for statistics. Additionally, we create a CL benchmark using the popular *GLUE classification* tasks (Wang et al., 2018) consisting of more than 5k train samples. We use the official validation split as test data and use $0.1\%$ of the train data to create a validation set. Our final benchmark includes five tasks; MNLI (353k/39k/9.8k), QQP (327k/36k/40k), QNLI (94k/10k/5.4k), SST-2 (60k/6.7k/872), CoLA (7.6k/856/1k). For vision experiments, we follow SupSup and use three CL benchmarks, SplitMNIST (Zenke et al., 2017) SplitCIFAR100 (Chaudhry et al., 2018) , and SplitTinyImageNet (Buzzega et al., 2020) datasets with 10, 100 and 200 total classes respectively.

**Metrics:** We follow Chaudhry et al. (2018) and evaluate our model after learning task $t$ on all the tasks, denoted by $\mathcal{T}$. This gives us an accuracy matrix $A \in \mathbb{R}^{n \times n}$, where $a_{i,j}$ represents the classification accuracy on task $j$ after learning task $i$. We want the model to perform well on all the tasks it has been learned. This is measured by the *average accuracy*, $\mathcal{A}(\mathcal{T}) = \frac{1}{N} \sum_{k=1}^{N} a_{N,k}$, where $N$ is the number of tasks. Next, we want the model to retain performance on the previous tasks when learning multiple tasks. This is measured by the *forgetting metric* (Lopez-Paz & Ranzato, 2017), $F(\mathcal{T}) = \frac{1}{N-1} \sum_{t=1}^{N-1} (\max_{k \in \{1,\ldots,N-1\}} a_{k,t} - a_{N,t})$. This is the average difference between the maximum accuracy obtained for task $t$ and its final accuracy. Higher accuracy and lower forgetting are desired.

| Method (↓) | GLUE | WebNLP | | | | |
|---|---|---|---|---|---|---|
| Order (→) | S1 | S2 | S3 | S4 | S5 | Average |
| *Random* | *33.3 (-)* | *7.14 (-)* | *7.14 (-)* | *7.14 (-)* | *7.14 (-)* | *7.14 (-)* |
| *Multitask* | *79.9 (0.0)* | *77.2 (0.0)* | *77.2 (0.0)* | *77.2 (0.0)* | *77.2 (0.0)* | *77.2 (0.0)* |
| *Individual* | *87.7 (0.0)* | *79.5 (0.0)* | *79.5 (0.0)* | *79.5 (0.0)* | *79.5 (0.0)* | *79.5 (0.0)* |
| FT | 14.1 (86.0) | 26.9 (62.1) | 22.8 (67.6) | 30.6 (55.9) | 15.6 (76.8) | 24.0 (65.6) |
| AdaptBERT + FT | 24.7 (53.4) | 20.8 (68.4) | 19.1 (70.9) | 23.6 (64.5) | 14.6 (76.0) | 19.6 (70.0) |
| AdaptBERT + Replay | 76.8 (3.8) | 73.2 (3.0) | 74.5 (2.0) | 74.5 (2.0) | 74.6 (2.0) | 74.2 (2.3) |
| MultiAdaptBERT | 78.5 (0.0) | 76.7 (0.0) | 76.7 (0.0) | 76.7 (0.0) | 76.7 (0.0) | 76.7 (0.0) |
| Prompt Tuning | 76.3 (0.0) | 66.3 (0.0) | 66.3 (0.0) | 66.3 (0.0) | 66.3 (0.0) | 66.3 (0.0) |
| Regularization | 72.5 (8.8) | 76.0 (2.8) | 74.9 (3.8) | 76.4 (1.8) | 76.5 (2.0) | 76.0 (2.6) |
| Replay | 77.7 (4.8) | 75.1 (3.1) | 74.6 (3.5) | 75.2 (2.2) | 75.7 (3.1) | 75.1 (3.0) |
| MBPA++† | - | 74.9 (-) | 73.1 (-) | 74.9 (-) | 74.1 (-) | 74.3 (-) |
| LAMOL† | - | 76.1 (-) | 76.1 (-) | 77.2 (-) | 76.7 (-) | 76.5 (-) |
| IDBR | 73.0 (6.8) | 75.9 (2.7) | 75.4 (3.5) | 76.5 (1.6) | 76.4 (1.9) | 76.0 (2.4) |
| SupSup | 78.3 (0.0) | 75.9 (0.0) | 76.1 (0.0) | 76.0 (0.0) | 75.9 (0.0) | 76.0 (0.0) |
| SSNᴇᴛ | 78.4 (3.6) | 76.3 (0.8) | 76.3 (0.8) | 76.4 (0.3) | 76.3 (0.3) | 76.3 (0.6) |
| ExSSNᴇᴛ | **80.5 (0.0)** | **77.0 (0.0)** | **77.1 (0.0)** | 76.7 (0.0) | **76.9 (0.0)** | **76.9 (0.0)** |

**Table 1:** Comparing average test accuracy ↑ (**and forgetting metric** ↓) for multiple tasks and sequence orders with state-of-the-art (SotA) methods. Results with † are taken from (Huang et al., 2021).

**Sparse Overlap to Quantify Parameter Interference:** Next, we propose a measure to quantify parameter interference for a task $i$, i.e., the fraction of the parameters in mask $M_i$ that are already updated by some previous task. We define *sparse overlap* as the difference between the number of parameters selected by mask $M_i$ and $M_i^{free}$ divided by the total parameters selected by $M_i$. Formally, we define *sparse overlap* (SO) between current supermask $M_i$ and supermasks for previous tasks $\{M_j\}_{j=1}^{i-1}$ as,

$$\text{SO}(M_i, \{M_j\}_{j=1}^{i-1}) = \frac{sum(M_i) - sum(M_i^{free})}{sum(M_i)} \ ; \ M_i^{free} = M_i \wedge \neg(\vee_{j=1}^{i-1}(M_j)) \tag{1}$$

where $\wedge, \vee, \neg$ are logical *and*, *or*, and *not* operations.

**Previous Methods and Baselines:** For both vision and language (**VL**) tasks, we compare with: **(VL.1) Naive Training/Finetuning** (Yogatama et al., 2019): where for the vision domain we train all model parameters from scratch whereas for the language domain we finetune BERT model weights for each task sequentially. **(VL.2) Experience Replay (ER)** (de Masson d'Autume et al., 2019): we replay previous tasks examples when we train new tasks; **(VL.3) Multitask Learning** (Crawshaw, 2020): where all the tasks are used jointly to train the model and have strong performance; **(VL.4) Individual Models**: where we train a separate model for each task. This is considered an upper bound for CL; **(VL.5) Supsup** (Wortsman et al., 2020). For natural language (**L**), we further compare with the following methods: **(L.6) Regularization** (Huang et al., 2021): Along with the Replay method, we regularize the hidden states of the BERT classifier with an L2 loss term; We show three Adapter BERT (Houlsby et al., 2019) variants, **(L.7) AdaptBERT + FT** where we have single adapter which is finetuned for all task; **(L.8) AdaptBERT + ER** where a single adapter is finetuned with replay; **(L.9) Multi-AdaptBERT** where a separate adapter is finetuned for each task; **(L.10) Prompt Tuning** (Li & Liang, 2021) that learns 50 different continuous prompt tokens for each task. **(L.11) MBPA++** (de Masson d'Autume et al., 2019) perform replay with random examples during training and does local adaptation during inference to select replay example; **(L.12) LAMOL** (Sun et al., 2019) uses a language model to generate pseudo-samples for previous tasks for replay; **(L.13) IDBR** (Huang et al., 2021) disentangles hidden representations into generic and task-specific representations and regularizes them while also performing replay. For vision task (**V**), we additionally compare with two popular regularization based methods, **(V.6) Online EWC** (Schwarz et al., 2018), **(V.7) Synaptic Intelligence (SI)** (Zenke et al., 2017); one knowledge distillation method, **(V.8) Learning without Forgetting (LwF)** (Li & Hoiem, 2017), three additional experience replay method, **(V.9) AGEM** (Chaudhry et al., 2018), **(V.10) Dark Experience Replay (DER)** (Buzzega et al., 2020), **(V.11) DER++** (Buzzega et al., 2020).

**Implementation Details:** We focus on the task incremental setting (Hsu et al., 2018) and unless otherwise specified, we obtain supermasks with a mask density of 0.1. For language tasks, unless specified otherwise we initialize the token embedding for our methods using a frozen BERT-base-uncased (Devlin et al., 2018) model's representations using Huggingface (Wolf et al., 2020). We use a static CNN model from Kim (2014) as our text classifier. Following Sun et al. (2019); Huang et al.

| Method | S-MNIST | S-CIFAR100 | S-TinyImageNet |
|---|---|---|---|
| *Multitask* | *96.5 (0.0)* | *53.0 (0.0)* | *45.9 (0.0)* |
| *Individual* | *99.7 (0.0)* | *75.5 (0.0)* | *53.7 (0.0)* |
| Naive Sequential | 49.6 (25.0) | 19.3 (73.7) | 11.5 (43.9) |
| EWC | 96.1 (4.5) | 32.4 (60.5) | 20.5 (52.1) |
| SI | 99.2 (0.6) | 46.1 (47.8) | 19.5 (46.2) |
| LwF | 99.2 (0.8) | 29.5 (70.2) | 18.1 (56.5) |
| AGEM | 98.3 (1.9) | 52.1 (42.0) | 21.6 (54.9) |
| ER | 99.2 (0.6) | 60.1 (27.5) | 35.6 (36.0) |
| DER | 98.9 (1.2) | 62.5 (28.4) | 35.9 (37.7) |
| DER++ | 98.3 (1.8) | 62.5 (27.5) | 36.2 (35.7) |
| SupSup | 99.6 (0.0) | 62.1 (0.0) | 50.6 (0.0) |
| SSNET | **99.7 (0.0)** | 23.9 (54.4) | 49.6 (1.9) |
| EXSSNET | **99.7 (0.0)** | **67.3 (0.0)** | **52.0 (0.0)** |

**Table 2:** Average accuracy ↑ **(Forgetting metric ↓)** on all tasks for vision. For our method we report the results are averaged over three random seeds.

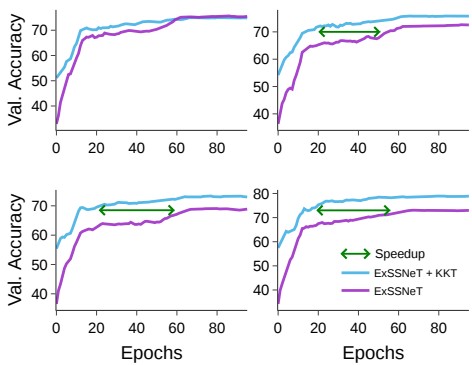

**Figure 4:** We plot validation accuracy vs Epoch for EXSSNET and EXSSNET + KKT. We observe that KKT helps to learn the subsequent tasks faster and improves performance.

(2021), for WebNLP datasets we learn different task orders S1-S5[2] that are provided in Appendix Table 6. Following (Wortsman et al., 2020), we use LeNet (Lecun et al., 1998) for SplitMNIST, a Resnet-18 model (Wortsman et al., 2020) for SplitCIFAR100, a ResNet50 model (He et al., 2016) for TinyImageNet datasets. Unless specified, we randomly split all the vision datasets to obtain five tasks with disjoint classes. In all our experiments, all methods perform an equal number of epochs over the datasets. For additional details on implementation and hyperparameters refer to Appendix A.1.1.

## 4.2 MAIN RESULTS

**Q1. Does Supermask Subnetwork Training Help?**

In these experiments, we show that EXSSNET outperforms multiple strong baseline methods including SupSup. For our main language experiments in Table 1, we sequentially learn multiple task orders, S1 - S5[2] corresponding to the GLUE and WebNLP benchmarks. These task orders are listed in Appendix Table 6. We report the average test accuracy (and forgetting in parentheses). For natural language, we perform better than previous state-of-the-art CL methods in four out of five cases, across multiple task orders, and in aggregate. Specifically, on the GLUE benchmark, EXSSNET is at least 2.0% better than other methods while avoiding CF. Furthermore, EXSSNET either outperforms or is close to the performance of the multitasking baseline which is a strong baseline for CL methods.

For vision tasks, we split the MNIST, CIFAR100, and TinyImageNet datasets into *five different tasks* with an equal number of disjoint classes and report results. We report these results in Table 2 and observe that EXSSNET leads to a 4.8% and 1.4% improvement over the strongest baseline for Split-CIFAR100 and Split-TinyImageNet datasets. Furthermore, both EXSSNET and SupSup outperform the multitask baseline. Moreover, EXSSNET bridges the gap to individually trained models significantly, for TinyImageNet we reach within 1.7% of individual models' performance. The average sparse overlap of EXSSNET is 19.4% across all three datasets implying that there is a lot more capacity in the model. See appendix Table 8 for sparse overlap of other methods.

We note that past methods employ specific tricks like local adaptation in MBPA++, a generative replay module in LAMOL, and experience replay in AGEM, DER, and ER. In contrast, EXSSNET does not require replay and simply trains different subnetworks for tasks.

**Q2. Can KKT Knowledge Transfer Module Share Knowledge Effectively?**

In Section 3.2, we presented our KKT module for enabling knowledge sharing across tasks. In Table 3, we show that adding the KKT module to EXSSNET, SSNET, and SupSup improves performance on vision benchmarks. The experimental setting here is similar to Table 2. We observe across all methods and datasets that the KKT module improves average test accuracy. Specifically, for the Split-CIFAR100 dataset, the KKT module results in 5.0%, and 3.2% improvement for SupSup

---

[2]For example, in S2 order the model learns the task in this order, ag → yelp → amazon → yahoo → dbpedia

| Method | S-MNIST | S-CIFAR100 | S-TinyImageNet |
|---|---|---|---|
| **SupSup** | 99.6 | 62.1 | 50.6 |
| **+ KKT** | 99.6 [+0.0] | **67.1** [+5.0] | **53.3** [+2.7] |
| **SSNet** | **99.7** | **23.9** | 49.6 |
| **+ KKT** | 99.3 [-0.4] | 23.5 [-0.4] | **51.8** [+2.2] |
| **ExSSNet** | 99.7 | 67.3 | 52.0 |
| **+ KKT** | 99.7 [+0.0] | **70.5** [+3.2] | **54.0** [+2.0] |

**Table 3:** We report average test accuracies ↑ **[and gains from KKT]** when using the KKT knowledge sharing module for Vision datasets. The overall best method is highlighted in gray.

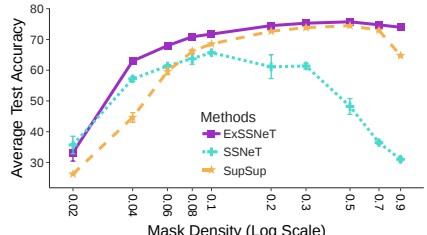

**Figure 5:** Average test accuracy versus mask density on SplitCIFAR100 dataset.

| Method | S-TinyImageNet | Avg. Sparse Overlap |
|---|---|---|
| **SupSup** | 90.34 (0.0) | 90.1 |
| **SSNet** | 89.02 (2.2) | 90.0 |
| **ExSSNet** | **91.21 (0.0)** | 90.0 |

**Table 4:** Average accuracy ↑ (**forgetting metric ↓**) and average sparse overlap when learning 100 tasks.

| Method | FastText | Glove | BERT |
|---|---|---|---|
| **SupSup** | 54.01 | 55.52 | 74.0 |
| **SSNet** | 60.41 [+6.4] | 59.78 [+4.3] | 74.5 [+0.5] |
| **ExSSNet** | **62.52** [+8.5] | **62.81** [+7.3] | **74.8** [+0.8] |

**Table 5:** Ablation result for token embeddings. We report average accuracy ↑ **[and gains over SupSup]**

and ExSSNet respectively; while for Split-TinyImageNet, ExSSNet + KKT outperforms the individual models. We observe a performance decline for SSNet when using KKT because KKT promotes sharing of parameters across tasks which can lead to worse performance for SSNet. Furthermore, ExSSNet + KKT outperforms all other methods on both the Split-CIFAR100 and Split-TinyImageNet datasets. For ExSSNet + KKT, the average sparse overlap is 49.6% across all three datasets (see appendix Table 8). These results indicate that the benefits of weight training and the KKT module can be coupled together to further improve performance.

**Q3. Can KKT Knowledge Transfer Module Improve Learning Speed of Subsequent Tasks?**

Next, we show that the KKT module enables us to learn new tasks faster. To demonstrate this, in Figure 4 we plot the running mean of the validation accuracy vs epochs for different tasks from the Split-CIFAR100 experiment in Table 3. We show curves for ExSSNet with and without the KKT module and omit the first task as both these methods are identical for Task 1 because there is no previous task to transfer knowledge. For all the subsequent tasks (Task 2,3,4,5), we observe that – (1) ExSSNet + KKT starts off with a much better initial performance compared to ExSSNet (2) given a fixed number of epochs for training, ExSSNet + KKT always learns the task better because it has a better accuracy at all epochs; and (3) ExSSNet + KKT can achieve similar performance as ExSSNet in much fewer epochs as shown by the green horizontal arrows. This clearly illustrates that using the KKT knowledge-transfer module not only helps to learn the tasks better (see Table 3) but it also learns them faster.

### 4.3 Additional Results and Analysis

**Q4. Effect of Mask Density on Performance:** Next, we show the advantage of using ExSSNet when the mask density is low. In Figure 5, we show the average accuracy for the Split-CIFAR100 dataset as a function of mask density. We observe that ExSSNet obtains 7.9%, 18.4%, 8.4%, and 4.7% improvement over SupSup for mask density values 0.02, 0.04, 0.06, 0.08 respectively. This is a highly desirable property as tasks select fewer parameters which inherently reduces sparse overlap allowing ExSSNet to learn a large number of tasks.

**Q5. Can ExSSNet Learn a Large Number of Tasks?** SupSup showed that it can scale to a large number of tasks. Next, we show that this property is preserved by ExSSNet. We perform experiments to learn 100 tasks created by splitting the TinyImageNet dataset. Note that, as the number of task increase, the sparse overlap between the masks also increases resulting in fewer free model weights. In the extreme case where there are no free weights, ExSSNet by design reduces to SupSup because there will be no weight training. From Table 4, we conclude that ExSSNet can learn 100 tasks while still improving performance over SupSup and preventing forgetting.

**Q6. Effect of Token Embedding Initialization for NLP:** For our language experiments, we use a pretrained BERT model (Devlin et al., 2019) to obtain the initial token representations. Using the powerful representations from a multitask model like BERT overshadows the superiority of

ExSSNeT over SupSup in terms of performance. Hence, we perform ablations on the token embedding initialization to further highlight these differences. In Table 5, we present the average test accuracy on the S2[2] task-order sequence of the sampled version of WebNLP dataset (see Section 4.1, Datasets). We initialize the token representations using *FastText* (Bojanowski et al., 2016), *Glove* (Pennington et al., 2014), and *BERT* embeddings. From Table 5, we observe that – (1) the performance gap between ExSSNeT and SupSup increases from $0.8\% \rightarrow 7.3\%$ and $0.8\% \rightarrow 8.5\%$ when moving from BERT to Glove and FastText initializations respectively. These gains imply that it is even more beneficial to use ExSSNeT in absence of good initial representations, and (2) the performance trend, ExSSNeT > SSNeT > SupSup is consistent across initialization.

## 5 RELATED WORK

Continual Learning methods fall into three main categories: *Regularization*, *Replay*, and *Architecture* based methods. We point the readers to Delange et al. (2021); Biesialska et al. (2020) for a comprehensive survey of all continual learning methods. Next, we discuss these main categories.

**Regularization-based methods** estimate the importance of model components and add importance regularization terms to the loss function. Zenke et al. (2017) regularize based on the distance of weights from their initialization, whereas Kirkpatrick et al. (2017); Schwarz et al. (2018) use an approximation of the Fisher information matrix (Pascanu & Bengio, 2013) to regularize the parameters. In NLP, regularization methods (Han et al., 2020; Wang et al., 2019) are used to constrain the relevant information from the huge amount of knowledge inside large language models (LLM). Huang et al. (2021) first identifying hidden spaces that need to be updated versus retained via information disentanglement (Fu et al., 2017; Li et al., 2020) and then regularize these hidden spaces separately. Our method inherently avoids the need for regularization as it avoids parameter interference.

**Replay based methods** maintain a small memory buffer of data samples (De Lange et al., 2019) or their relevant proxies (Rebuffi et al., 2017) from the previous tasks and retrain for them later to prevent CF. Lopez-Paz & Ranzato (2017); Chaudhry et al. (2018) use the buffer during optimization to constrain parameter gradients. Shin et al. (2017); Kemker & Kanan (2018) uses a generative model to sample and replay pseudo-data during training, whereas Rebuffi et al. (2017) replay distilled knowledge from the previous tasks. de Masson d'Autume et al. (2019) employ episodic memory along with local adaptation for CL in the NLP domain, whereas Sun et al. (2019) trains a language model to generate a pseudo-sample for replay. These methods usually have associated memory and runtime costs whereas our method works without a replay buffer.

**Architecture based methods** can be divided into two categories: (1) methods that add new modules over time (Sodhani et al., 2019; Veniat et al., 2021; Douillard et al., 2022); and (2) methods that isolate the network's parameters for different tasks (Fernando et al., 2017; Mallya & Lazebnik, 2018; Mallya et al., 2018). Rusu et al. (2016) introduce a new network for each task that is connected to all the previous tasks resulting in super-linear growth in network size. Schwarz et al. (2018) fix this issue by distilling the new network after each task into the original one. Recent prompt learning-based CL models for vision (Wang et al., 2022a;b) assume access to a pre-trained model to learn a set of prompts that can potentially be shared across tasks to perform CL this is orthogonal to our method that trains from scratch. Methods like Fernando et al. (2017) initialize a fixed-size model and reuse a subset of modules for each task by finding a path in the graph of neural network modules. Mallya & Lazebnik (2018) allocates parameters to specific tasks and then trains them in isolation which limits the number of tasks that can be learned. In contrast, Mallya et al. (2018) use a frozen pretrained model and learns a new mask for each task but a pretrained model is crucial for their method's good performance. Wortsman et al. (2020) removes the pretrained model dependence and learns a mask for each task over a fixed randomly initialized network. ExSSNeT avoids the shortcomings of Mallya & Lazebnik (2018); Mallya et al. (2018) and performs supermask subnetwork training to increase the representational capacity compared to (Wortsman et al., 2020) while performing knowledge transfer and avoiding CF.

## 6 DISCUSSION AND CONCLUSION

In this paper, we define a simple yet effective method ExSSNeT and the KKT knowledge transfer module that leverages sparsity to learning tasks while resolving the shortcomings of supermasks by performing Exclusive Subnetwork Training. ExSSNeT improves performance, prevents forgetting, accumulates and transfers knowledge across tasks, and works for both vision and language domains.

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

# A  APPENDIX FOR EXSSNET

## A.1  ADDITIONAL EXPERIMENTS AND DETAILS

| ID | Task Sequence |
|----|---------------|
| S1 | mnli → qqp → qnli → sst2 → cola (Dec. data Size) |
| S2 | ag → yelp → amazon → yahoo → dbpedia |
| S3 | yelp → yahoo → amazon → dbpedia → ag |
| S4 | dbpedia → yahoo → ag → amazon → yelp |
| S5 | yelp → ag → dbpedia → amazon → yahoo |
| S6 | ag → yelp → yahoo |
| S7 | yelp → yahoo → ag |
| S8 | yahoo → ag → yelp |

**Table 6:** Task sequences used in text experiments. For the GLUE dataset, we use order corresponding to decreasing train data size. Sequence S2-S8 are from (Huang et al., 2021; de Masson d'Autume et al., 2019; Sun et al., 2019).

| Dataset | Class | Type | Train | Validation | Test |
|---------|-------|------|-------|------------|------|
| AGNews | 4 | News | 8k | 8k | 7.6k |
| Yelp | 5 | Sentiment | 10k | 10k | 7.6k |
| Amazon | 5 | Sentiment | 10k | 10k | 7.6k |
| DBPedia | 14 | Wikipedia | 28k | 28k | 7.6k |
| Yahoo | 10 | Q&A | 20k | 20k | 7.6k |

**Table 7:** Statistics for sampled data used from Huang et al. (2021) for hyperparameter tuning. The validation set is the same size as the train set. Class means the number of output classes for the text classification task. Type is the domain of text classification.

### A.1.1  EXPERIMENTAL SETUP AND HYPERPARAMETERS

**Implementation Details:** Unless otherwise specified, we obtain supermasks with a mask density of 0.1. In our CNN models, we use non-affine batch normalization to avoid storing their means and variance parameters for all tasks (Wortsman et al., 2020). Similar to (Wortsman et al., 2020), bias terms in our model are 0 and we randomly initialize the model parameters using *signed kaiming constant* (Ramanujan et al., 2019). We use Adam optimizer (Kingma & Ba, 2014) along with cosine decay (Loshchilov & Hutter, 2016) and conduct our experiments on GPUs with 12GB of memory. For our main experiment, we run three independent runs for each experiment and report the averages for all the metrics and experiments. For natural language tasks, unless specified otherwise we initialize the token embedding for our methods using a frozen BERT-base-uncased (Devlin et al., 2018) model's representations using Huggingface (Wolf et al., 2020). We use a static CNN model from Kim (2014) as our text classifier over BERT representations. The model employs 1D convolutions along with *Tanh* activation. Following Sun et al. (2019); Huang et al. (2021), we evaluate our model on various task sequences as provided in Appendix Table 6, while limiting the maximum number of tokens to 256. Following (Wortsman et al., 2020), we use LeNet (Lecun et al., 1998) for SplitMNIST dataset, a Resnet-18 model with fewer channels (Wortsman et al., 2020) for SplitCIFAR100 dataset, a ResNet50 model (He et al., 2016) for TinyImageNet dataset. Unless specified, we randomly split all the vision datasets to obtain five tasks with disjoint classes. We use the codebase of DER (Buzzega et al., 2020) to obtain the vision baselines. In all our experiments, all methods perform an equal number of epochs over the datasets.

For the ablation experiment on natural language data, following Huang et al. (2021), we use a sampled version of the WebNLP datasets due to limited resources. The reduced dataset contains 2000 training and validation examples from each output class. The test set is the same as the main experiments. The dataset statistics are summarized in Table 7. For WebNLP datasets, we tune the learning rate

**Table 8:** We report the average sparse overlap for all method and dataset combinations reported in Table 3.

| Method | S-MNIST | S-CIFAR100 | S-TinyImageNet |
|---|---|---|---|
| **SupSup** | 22.6 | 18.9 | 18.1 |
| **+ KKT** | 46.4 | 48.3 | 52.4 |
| **SSNET** | 22.5 | 17.6 | 18.6 |
| **+ KKT** | 52.7 | 49.9 | 52.4 |
| **ExSSNET** | 22.5 | 17.3 | 18.5 |
| **+ KKT** | 47.8 | 48.8 | 52.4 |

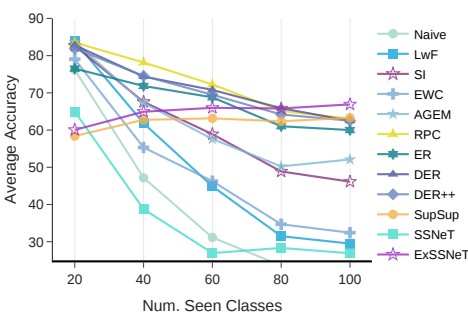

**Figure 6:** Average Accuracy of all seen tasks as a function of the number of learned classes for the Split-CIFAR100 dataset.

on the validation set across the values $\{0.01, 0.001, 0.0001\}$, for GLUE datasets we use the default learning rate of the BERT model. For our vision experiments, we use the default learning rate for the dataset provided in their original implementation. For TinyImageNet, SplitCIFAR100, SplitMNIST dataset, we run for 30, 100, and 30 epochs respectively. We store 0.1% of our vision datasets for replay while for our language experiments we use 0.01% of the data because of the large number of datasets available for them.

### A.1.2 SPARSE OVERLAP NUMBERS

In Table 8, we report the sparse overlap numbers for SupSup, SSNET, and ExSSNET with and without the KKT knowledge transfer module. This table corresponds to the results in main paper Table 3.

### A.1.3 AVERAGE ACCURACY EVOLUTION

In Figure 6, we plot $\sum_{i \leq t} A_{ti}$ vs $t$, that is the average accuracy as a function of observed classes. This plot corresponds to the SplitCIFAR100 results provided in the main paper Table 2. We can observe from these results that Supsup and ExSSNeT performance does not degrade when we learn new tasks leading to a very stable curve whereas for other methods the performance degrades as we learn new tasks indicating some degree of forgetting.

d

| Model (↓) | Length-5 WebNLP | | | | Length-3 WebNLP | | | |
|---|---|---|---|---|---|---|---|---|
| Order (→) | S2 | S3 | S4 | Average | S6 | S7 | S8 | Average |
| *Random* | *7.14* | *7.14* | *7.14* | *7.14* | *10.0* | *10.0* | *10.0* | *10.0* |
| *MTL* | *75.09* | *75.09* | *75.09* | *75.09* | *74.16* | *74.16* | *74.16* | *74.16* |
| **Finetune**† | 32.37 | 32.22 | 26.44 | 30.34 | 25.79 | 36.56 | 41.01 | 34.45 |
| **Replay**† | 68.25 | 70.52 | 70.24 | 69.67 | 69.32 | 70.25 | 71.31 | 70.29 |
| **Regularization**† | 72.28 | 73.03 | 72.92 | 72.74 | 71.50 | 70.88 | 72.93 | 71.77 |
| **AdaptBERT** | 30.49 | 20.16 | 23.01 | 24.55 | 24.48 | 31.08 | 26.67 | 27.41 |
| **AdaptBERT + Replay** | 69.30 | 67.91 | 71.98 | 69.73 | 66.12 | 69.15 | 71.62 | 68.96 |
| **IDBR**† | 72.63 | 73.72 | 73.23 | 73.19 | 71.80 | 72.72 | 73.08 | 72.53 |
| **SupSup** | 74.01 | 74.04 | 74.18 | 74.08 | 72.01 | 72.35 | 72.53 | 72.29 |
| **SSNET** | 74.5 | 74.5 | 74.65 | 74.55 | 73.1 | 72.92 | 73.07 | 73.03 |
| **ExSSNET** | **74.78** | **74.72** | **74.71** | **74.73** | **72.67** | **72.99** | **73.24** | **72.97** |

**Table 9:** Average test accuracy reported over task sequences for three independent runs on sub-sampled data. Results with † are taken from Huang et al. (2021).

| Method | Runtime (in minutes) |
|---|---|
| *Multitask* | 200 |
| **Finetune** | 175 |
| **Replay** | 204 |
| **AdapterBERT + FT** | 170 |
| **AdapterBERT + Replay** | 173 |
| **MultiAdaptBERT** | 170 |
| **Regularization** | 257 |
| **IDBR** | 258 |
| **SupSup** | **117** |
| **SSNET** | **117** |
| **EXSSNET** | **117** |

**Table 10:** Runtime comparison of different methods used in the text experiments.

| Method (↓) | GLUE | WebNLP | | | | |
|---|---|---|---|---|---|---|
| Order (→) | S1 | S2 | S3 | S4 | S5 | Average |
| *Random* | *33.3* | *7.14* | *7.14* | *7.14* | *7.14* | *7.14* |
| *Multitask* | *80.6* | *77.4* | *77.5* | *76.9* | *76.8* | *77.1* |
| **FT** | 14.0 | 27.0 | 22.9 | 30.4 | 15.6 | 24.0 |
| **Replay** | 79.7 | 75.2 | 74.5 | 75.2 | 75.5 | 75.1 |
| **AdaptBERT + FT** | 25.1 | 20.8 | 19.1 | 23.6 | 14.6 | 19.5 |
| **AdaptBERT + Replay** | 78.6 | 73.3 | 74.3 | 74.7 | 74.6 | 74.2 |
| **MultiAdaptBERT** | 83.6 | 76.7 | 76.7 | 76.7 | 76.7 | 76.7 |
| **Regularization** | 75.5 | 75.9 | 75.0 | 76.5 | 76.3 | 75.9 |
| **IDBR** | 77.5 | 75.8 | 75.4 | 76.4 | 76.4 | 76.0 |
| **SupSup** | 78.1 | 75.7 | 76.0 | 76.0 | 75.9 | 75.9 |
| **SSNET** | 77.2 | 76.3 | 76.3 | 77.0 | 76.1 | 76.4 |
| **EXSSNET** | 80.1 | 77.1 | 77.3 | 77.2 | 77.1 | 77.2 |

**Table 11:** Average validation accuracy (↑) for multiple tasks and sequence orders with previous state-of-the-art (SotA) methods.

### A.1.4 EFFECT OF TASK ORDER AND NUMBER OF TASKS

Following Huang et al. (2021), we conduct experiments to study the effect of task length and order in the language domain. We use task sequences of lengths three and five, with multiple different task orders on the sampled data (Section 4.1, Table 6, and Appendix) to characterize the impact of these variables on the performance. In Table 9, we present the average test accuracy averaged over three different random seeds. We observe that across all six different settings our method performs better compared to all the baseline methods. Our methods bridge the gap toward multitask methods' performance, leaving a gap of 0.36% and 1.19% for lengths three and five sequences, respectively.

### A.1.5 RUNTIME COMPARISON ACROSS METHODS

In this Section, we provide the result to compare the runtime of various methods used in the paper. We ran each method on the sampled version of the WebNLP dataset for the *S2* task order as defined in Table 6. We report the runtime of methods for four epochs over each dataset in Table 10. Note that the masking-based method, SupSup, SSNET, EXSSNET takes much lower time because they are not updating the BERT parameters and are just finding a mask over a much smaller CNN-based classification model using pretrained representation from BERT. This gives our method an inherent advantage that we are able to improve performance but with significantly lower runtime while learning a mask over much fewer parameters for the natural language setting.

### A.1.6 VALIDATION RESULTS

In Table 11, we provide the average validation accuracies for the main natural language results presented in Table 1. We do not provide the validation results of LAMOL (Sun et al., 2019) and MBPA++ (de Masson d'Autume et al., 2019) as we used the results provided in their original papers. For the vision domain, we did not use a validation set because no hyperparameter tuning was performed as we used the experimental setting and default parameters from the original source code from (Wortsman et al., 2020; Wen et al., 2020).

## A.2 ADDITIONAL MODEL DETAILS

### A.2.1 ALGORITHM FOR EXSSNET

In Algorithm 1, we provide a pseudo-code for our method EXSSNET for easier reference and understanding. We also attach our working code as supplementary material to encourage reproducibility.

---

**Algorithm 1** EXSSNET training procedure.

---

**Input:** Tasks $\mathcal{T}$, a model $\mathcal{M}$, mask sparsity $k$, exclusive=True
**Output:** Trained model
▷ Initialize model weights $W^{(0)}$
initialize_model_weights($\mathcal{M}$)
**forall** $i \in range(|\mathcal{T}|)$ **do**
    ▷ Set the mask $M_i$ corresponding to task $t_i$ for optimization.
    mask_opt_params = $M_i$
    ▷ Learn the supermask $M_i$ using edge-popup
    **forall** $em \in mask\_epochs$ **do**
     |  $M_i$ = learn_supermask(model, mask_opt_params, $t_i$)
    **end**
    ▷ Model weight at this point are same as the last iteration $W^{(i-1)}$
    **if** $i > 1$ *and exclusive* **then**
       ▷ Find mask for all the weights used by previous tasks.
       $M_{1:i-1} = \vee_{j=1}^{i-1}(M_j)$
       ▷ Get mask for weights in $M_i$ which are not in $\{M_i\}_{j=1}^{i-1}$
       $M_i^{free} = M_i \wedge \neg M_{1:i-1}$
       ▷ Find non-overlapping weight for updating.
       $W_{free}^{(i)} = M_i^{free} \odot W^{(i-1)}$
    **else if** *not exclusive* **then**
       $W_{free}^{(i)} = W^{(i-1)}$
    **end**
    weight_opt_params = $W_{free}^{(i)}$
    ▷ Learn the free weight in the supermask $M_i$
    **forall** $em \in weight\_epochs$ **do**
     |  $W^{(i)}$ = update_weights(model, weight_opt_params, $t_i$)
    **end**
**end**

---

### A.2.2 MODEL DIAGRAM FOR SUPSUP

In Figure 7, we provide the canonical model diagram for SupSup. Please read the figure description for more details regarding the distinctions between SupSup and ExSSNeT.

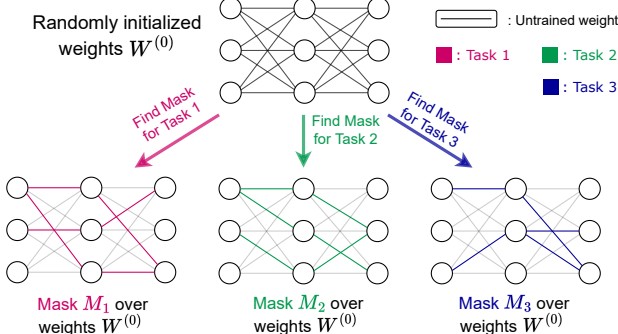

**Figure 7:** This is a canonical model diagram for SupSup. In SupSup, the model weights are always fixed at the random initialization $W^{(0)}$. For each task SupSup learns a new mask (in this case $M_1, M_2, M_3$) over the weights $W^{(0)}$. A mask selectively activates a subset of weights for a particular task. This subset of selected weights forms a subnetwork inside the full model which we refer to as the supermask subnetwork. For example, when learning Task 2, SupSup learns the mask $M_2$ (the weights activated by the mask are highlighted in green) over the fixed weight $W^{(0)}$. These highlighted weights along with the participating nodes are the subnetwork formed by mask $M_2$. Whenever a prediction is made for Task 2 samples, this mask is selected and used to obtain the predictions. Please note that the model weights $W^{(0)}$ are never updated after their random initialization. Hence, for SupSup there is no learned knowledge sharing across tasks. This is in contrast to our setup in Figure 1, where for the first task the mask is learned over the weights $W^{(0)}$ but once the mask is selected the weights of the corresponding subnetwork are also updates to obtain new weight $W^{(1)}$. Then the next task's mask is learned over these new set of weights $W^{(1)}$ and so on. Also note that in Figure 1, we do not show the KKT knowledge transfer module here to avoid confusion.

