# OpenReview forum: "Exclusive Supermask Subnetwork Training for Continual Learning"
_ICLR.cc/2023/Conference — Submitted to ICLR 2023_

### Official Review · Reviewer_jWtC · 2022-10-20

**Confidence:** 4
**Correctness:** 4
**Technical Novelty And Significance:** 2
**Empirical Novelty And Significance:** 2
**Recommendation:** 3

**Clarity, Quality, Novelty And Reproducibility:**

- -The technical contributions of the paper are quite limited. Indeed, the authors build on top of a previous work, SupSup, and add a few incremental updates to improve performance. The first improvement relies on allowing parameters of subnetworks to be updated by the optimization (and freezing the ones used by previous tasks). This contribution is very intuitive and effective, but its addition doesn't constitute sufficient novelty per se. Moreover, the model resembles a number of parameter isolation methods that were proposed in the last few years [1,2,3,4]. Only a couple of them are discussed and the authors claim ExSSNet avoids their shortcomings. However, none of these methods is compared against. More specifically, [1,2] rely on task-specific subnetworks, whose weights and masks are learned jointly, and have similar mechanisms to block gradient updates on parameters used in previous tasks. In this respect, differences between these methods and the proposed work is in details. Finally, the KKT component is interesting and effective, but its employment is a bit contextual to SupSup.
- -Section 2 is not necessary, as it reiterates the motivations of the paper in a more detailed way. In my opinion, motivations are convincing enough in the introduction.
- -Difference between SSNet and exSSNet does not emerge clearly in Figure 2. I however acknowledge it emerges in other experiments.
- -I struggle understanding the notation paragraph in section 3. Specifically, a node is defined as $\mathcal{Z}_v=\sigma(\mathcal{I}_v)$, where $\mathcal{I}_v$ are inputs, $\mathcal{Z}_v$ are outputs, and $\sigma$ is defined as an activation function. This way, it seems that a node is the same as an activation function.

[1] Serra, Joan, et al. "Overcoming catastrophic forgetting with hard attention to the task." International Conference on Machine Learning. PMLR, 2018.

[2] Abati, Davide, et al. "Conditional channel gated networks for task-aware continual learning." Proceedings of the IEEE/CVF Conference on Computer Vision and Pattern Recognition. 2020.

[3] Mallya, Arun, Dillon Davis, and Svetlana Lazebnik. "Piggyback: Adapting a single network to multiple tasks by learning to mask weights." Proceedings of the European Conference on Computer Vision (ECCV). 2018.

[4] Kim, Jangho, Jeesoo Kim, and Nojun Kwak. "StackNet: Stacking feature maps for Continual learning." Proceedings of the IEEE/CVF Conference on Computer Vision and Pattern Recognition Workshops. 2020.


**Strength And Weaknesses:**

+ +The paper is very clear and well written, one read suffices to the interested reader to grasp the main ideas and technical contributions.
+ +Experiments showcase encouraging results on two different data modalities and tasks, for a total of 5 different datasets among NLP and vision.
+ +Experimental results are very encouraging, as the model is able to outperform a number of recent works, including some relying on replay buffers.
+ +An ablation study grounds the effectiveness of KKT, grounding its beneficial effect.
- -The biggest weakness of the paper is its technical novelty, that I'll discuss in the next point.
- -The model, needing to separate the relevant task-specific subnetwork during inference, can only be employed in task-incremental learning settings, which represent the easiest and more controlled protocol in CL literature. This raises questions about the suitability of the model for real problems.

**Summary Of The Paper:**

This paper is proposing a model for continual learning based on subnetwork isolation and training. Specifically, the authors build on top of SupSup, a prior method relying on a randomly initialized backbone where task-specific subnetworks are discovered and isolated, showcasing good performances on a stream of tasks to be learned. This paper extends the method by i) allowing subnetwork parameters to be updated by gradient descent, and therefore achieving better performance on their task and ii) introducing a KNN-based mask initialization method for bootstrapping subnetwork selection on the next task. Experiments are carried out for natural language processing and computer vision tasks.

**Summary Of The Review:**

I appreciate the quality of the research and of the experimental section. However, in my opinion, the technical contribution of the paper is not worth for publication in a venue such as ICLR. I really think this work could benefit more discussion and comparisons with the parameter isolation methods I mentioned above.

---

> ### Author Response · Authors · 2022-11-15
> **Response to Reviewer jWtC (Part -1)**
>
> **Restriction to Task Incremental Setting:** Firstly, similar to SupSup, our idea could be extended to the case where the task identities are not provided during the inference. The SupSup paper presents a way to do this in their Section 3.3 and equation 4, where they propose to minimize entropy to select the best mask during inference. This can be directly applied to ExSSNeT in a setting where the task identities are not provided during inference. This lies outside the scope of our problem statement which was to improve the performance of supermasks for CL as also noted by you in the statement, “*The work is well motivated. It starts from the limitation of the SupSup method and designs well-motivated and well-justified techniques to improve the supermask-based method –Reviewer 7z7s*”
>
>
> **Value of Task Incremental Setting in NLP and Practical Applications:** Secondly, we would like to argue that this paradigm of known task identities is extremely popular in NLP because of the way these models are used in the current practical scenarios. There are significant challenges like knowledge sharing and scaling to many tasks in this case as well. Hence, we argue for the importance of this setting for practical use cases. Moreover, in Table 1, we compare with popular parameter isolation methods like prompt tuning and parameter-efficient adaptors in the setting of known task identities.
>
>
> **Novelty of our work:** Firstly, the value of our work comes from – 1) addressing some of the major issues in important past works which are highlighted in Section 1 paragraph three, 2) the concrete demonstration and quantification of the problems shown in Section 2, 3) proposing a simple but effective exclusive weight training based solution which works well, 4) our novel KKT module which is a simple and lightweight module to tackle the important problem of knowledge transfer, 5) Our thorough and well-designed experiments to demonstrate the utility and functionality of each of the proposed contribution, and 6) applicability of our method on both multiple domains.
>
>
> These contributions are also noted by other reviewers who find our idea to be novel and different enough compared to SupSup, as shown by these statements: 1) “*The idea of the overall framework is novel compared to previous work SupSup – Reviewer fPix*”, 2) *“The work is well motivated. It starts from the limitation of the SupSup method and designs well-motivated and well-justified techniques to improve the supermask based method. – Reviewer 7z7s” *, 3) “*The experiments are sufficient and can effectively validate the claimed points. – Reviewer fPix*”, and 4) “*The paper conducted carefully designed analyses and discussions on the experiments. – Reviewer 7z7s*”.
>
> The value of our work lies in the simplicity of the idea that is applicable to both text and vision domains with strong results and through analysis of the method and its performance. Hence, we believe our work will be of value to the research community as well as practitioners.
>
> **Comparison with Baselines:** In Table 1 for the NLP domain, we compared with multiple very recent parameter isolation methods like prompt tuning, and adaptors and showed that our method outperforms them. For the vision domain, we compare with the most relevant and well performant 8 baseline methods. We selected the most popular and performant methods from each category of regularization (EWC, SI), Gradient projection (A-Gem), replay (ER), replay+distillation (DER, DER++) based methods, and parameter isolation (SupSup). Hence, we skipped comparing with the plethora of CL methods in the literature. Additionally, please note that ExSSNeT even outperforms the replay methods which store and relearn from samples and are considered to be the strongest types of continual learning method. Hence, we believe there is enough evidence for our methods' strong performance and utility.
>
>
> **Generality of KKT module:** The main idea of KKT is to use KNN to efficiently estimate the predictive power of the subnetworks of the previously learned tasks. We argue that similar formulations could be used in multiple CL scenarios where tasks have additional parameters. For example, a similar formulation can be used to find the best adaptor weight / prompts from the previous tasks to initialize new task adaptors / prompt vectors to learn faster and better. Hence, the KKT module is not just limited to SupSup and can be applied to a broader category of CL methods with slight modifications.

---

> > ### Author Response · Authors · 2022-11-15
> > **Response to Reviewer jWtC (Part - 2)**
> >
> > **Different between SSNeT and ExSSNeT in Figure 2:** As stated in section 2 - Problem 1, the experiment In Figure 2 is on a “*single 100-way classification*”. In Figure 2, we show that if the selected subnetworks correspond to a supermask, then sparse weight training can match the performance of a densely trained model. Hence, in this case, SSNeT and EXSSNeT are the same models as there is a single task. Whereas, in Figure 3 we learn *“ten 10-way classification tasks” *which is a continual learning scenario where SSNeT shows catastrophic forgetting while  ExSSNeT does not.
> >
> > **Question about notation:** As stated in the notation section of the paper, “*For a node $v$ in layer $l$, let $I_v$ denote its input and $Z_v = σ(I_v)$ denote its output, where $σ(.)$ is the activation function*”. We would like to note that a node, a.k.a neuron in multilayer perceptron terminology, is not defined by $I_v$ or $Z_v$ or the activation function.
> >
> > We hope that we have addressed your main concerns and if so then please consider increasing your score. Thank you so much for your time!

---

> > > ### Comment · Reviewer_jWtC · 2022-11-22
> > > **Final recommendation**
> > >
> > > I thank the authors for their detailed response. However, I'd like to keep my initial reject score.
> > >
> > > As stated in my first review, I see the parameter masking presented here very similar to the ones in [1] and [2]. It is true that the solution improves over SupSup, but in my view it is a simple application of those ideas (which are widely known in CL community) to the SupSup model. In their response, the authors iterated over their contributions but without really commenting on this point.
> > >
> > > Moreover, the authors claim they select as baselines the most popular methods of each family. However, they lack parameter isolation methods like HAT, which are the most relevant for the presented model.
> > >
> > > Finally, I remark that outperforming buffer based models is remarkable in class-incremental settings. In task-incremental, there has already been some research highlighting architectural or parameter isolation models can outperfom memory buffers. For instance, in the DER paper, authors show PNNs outperform DER.

---

> > > > ### Author Response · Authors · 2022-11-27
> > > > **Comparisons with CGATE and differences with past works.**
> > > >
> > > > **Differences compared to HAT [1] and CGATE [2]:** First of all, we would like to point out that our method is a parameter isolation method and hence shares the high-level idea of segregating parameters with [1] and [2]. This is in general true for all the methods stemming from a particular chain of thought in research. Next, we note the differences in the low-level details between ExSSNeT compared to these methods. [2] uses channel gating to select some kernels for each task, in contrast to this we operate on a more fine-grained level of selecting individual weights from the networks, i.e. a single kernel can be shared by multiple tasks but only a few parameters from each kernel can be used for a task. We induce sparsity at the parameter level as opposed to selecting a few kernels. Moreover, the selection mechanism of our work is guided by finding supermasks that are known to be a good initialization that helps with better optimization of the task, whereas [2] uses a gating network to select the kernels.
> > > >
> > > > On the other hand, similar to [1], we also work with individual parameters but HAT [1] does not use a pure binary mask as mentioned in their paper – *“Another difference between HAT and the previous three approaches is that it does not use purely binary masks.”*, whereas in our method we have a binary selection of the parameters.
> > > >
> > > > Moreover, both HAT [1] and CGATE [2] do not address an important question of knowledge transfer from previously learned tasks to new tasks. In contrast, we address this challenge with our simple and novel knowledge-sharing KKT module that is general enough to be applied to any parameter isolation method and leads to a significant boost in performance as well as results in much faster learning.
> > > > Hence, these differences combined with our contribution mentioned in the previous comment make our work different from past works.
> > > >
> > > > **Comparison with parameter Isolation Method CGATE [2]:** To address your concern regarding comparisons with the parameter isolation method, in our original paper we have already compared with SupSup which is one of the strongest parameter isolation methods. However, to address your concern further, we additionally provide the results for CGATE [2] and find that our method **ExSSNeT is 7.2% better than CGATE** and **SupSup is 2% better than CGATE** on the Split-CIFAR100 dataset corresponding to Table 2 in the paper. We provide the results below as a table.
> > > >
> > > > | Method | S-MNIST | S-CIFAR100 |
> > > > |---|---|---|
> > > > | SupSup | 99.6 | 62.1 |
> > > > | CGATE [2] | 99.6 | 60.1 |
> > > > | ExSSNeT | **99.7** | **67.3** |
> > > >
> > > > We hope that this addresses some of your major concerns and would request you to re-evaluate the paper.

---

### Official Review · Reviewer_fPix · 2022-10-22

**Confidence:** 4
**Correctness:** 3
**Technical Novelty And Significance:** 3
**Empirical Novelty And Significance:** Not applicable
**Recommendation:** 6

**Clarity, Quality, Novelty And Reproducibility:**

The originality in terms of framework is good while the originality of specific method is not so creative. The overall quality and clarification of the work meets the standard of the conference.

**Strength And Weaknesses:**

Strength:
- The idea of the overall framework is novel compared to previous work SupSup. It is a natural idea to expend fixed weights to trainable weights but it brings a large boost and more advantages, e.g., under the condition of sparse masks.
- The experiments are sufficient and can effectively validate the claimed points. The informative figures and tables are also helpful for illustration and understanding.
- The paper is well organized and comprehensive in terms of experimental settings and details.

Weakness:
- Although the idea is novel, most of the specific implementation of the framework is based on existing work. For example, the supermask learning follows Ramanujan et al.(2019), the training mechanism using exclusive mask is similar as Learn to Grow[A].
- The description of the paper can be polished. Some long sentences interfere with reading, and there are also a few typos, e.g., the first sentence of the fourth paragraph, “We overcome the aforementioned issues, we propose our method,...”.
- The main concern is about the claim that ExSSNet can learn 100 tasks. First, as listed in Table 4, SupSup actually achieves a good results (90.34%) and the improvement of ExSSNet (91.21%) is not noticeable. Therefore, the ability to learn 100 tasks is what SupSup have achieved, but not unique for ExSSNet. Second, I wonder that this kind of ability is related to the size of the model since the extreme case is there are no free weights. How is the performance of using a small model like LeNet for SplitMNIST and ResNet18 for SplitCIFAR100. Furthermore, how is the performance of using a larger model like Resnet101? Can larger models learn more than 100 tasks since they have more parameter spaces?

Reference:
- [A] Xilai Li, Yingbo Zhou, Tianfu Wu, Richard Socher. "Learn to grow: A continual structure learning framework for overcoming catastrophic forgetting." International Conference on Machine Learning. PMLR, 2019.

**Summary Of The Paper:**

This paper proposes an Exclusive Supermask Subnetwork Training framework for continual learning of both text classification and vision tasks. Compared to the previous method SupSup, the proposed ExSSNet makes fixed weights trainable thus facilitating the knowledge transfer from previously learned tasks to new tasks. The further proposed KKT module can be regarded as a better Initialization mechanism, which helps to learn new tasks faster and better. Extensive experiments and impressing performance validate the effectiveness of the proposed method.

**Summary Of The Review:**

I think this paper can be regarded as an good improvements of the previous method SupSup thus I tend to accept. But my main concern is about the claim of learning a large number of tasks. If the author can give good feedback, I will improve the score.

---

> ### Author Response · Authors · 2022-11-15
> **Response to Reviewer fPix**
>
> **Learning a Large Number of Tasks:** We would like to note a few things here. First, the ability to learn multiple tasks is dependent on the complexity of the tasks and the ability of the model to handle catastrophic forgetting. In the case of TinyImageNet, there are 200 classes in total which are divided into 100 binary classification tasks which are easier to perform compared to multiway classification tasks. This limits the performance improvements ExSSNeT can achieve over SupSup. This is canonical to the case of the MNIST dataset where we perform binary classification and in that case, most methods have similar but good performance.
>
> **ImageNet Experiments:** To clarify this further, we provide additional experiments on learning 100 tasks from the ImageNet dataset where each tasks is a 10-way classification. We report the performance of both SupSup and ExSSNeT. We observe that there is a significant gap of 3% compared to the 0.84% improvement on 100 tasks from TinyImageNet.
>
> | Method | Average Accuracy |
> |---|---|
> | SupSup | 70.17 |
> | ExSSNeT | 73.21 |
>
>
> **Challenges in scaling ExSSNeT:** We agree with the point that Supsup scales to a large number of tasks because they don't train the weights and keep them fixed but when training weight to improve performance, it is nontrivial to learn multiple tasks. Hence, forgetting is a real challenge especially when learning large numbers of tasks, as stated in Section 1 paragraph 3, and Section 2 problem 2. This is true for SSNeT where we observe forgetting, in contrast for ExSSNeT, we prevent the issue of forgetting by exclusive weight training even when learning a large number of tasks. Hence, in the case of ExSSNeT, there are challenges to be overcome for it to scale to 100 tasks as compared to SupSup. Our soul purpose here is to show that, ExSSNeT preserves the scaling property of SupSup while resulting in some improvements.
>
>
> **Bigger Model Scale better:** as we start learning tasks, we initially have a lot of free parameters, and in these cases, ExSSNeT outperforms SupSup by a significant margin. This is implied from Figure 1, where we show that weight training for a single task helps. In the case when we are learning a really huge number of tasks, after a point there might be very few free parameters left and in these cases, our method becomes similar to SupSup. Hence, our model acts as an upper bound on the performance of SupSup even in the worst case. Moreover, as pointed out by you, if we use larger models there are more free parameters and hence our method will lead to even more improvement over SupSup. This is true by design and hence we skip these experiments in the paper.
>
> **Fixing Long Sentences and typos:** We promise to further polish the paper in the final version.
>
> We hope that we have addressed your main concerns and if so then please consider increasing your score. Thank you so much for your time and valuable comments!

---

> > ### Author Response · Authors · 2022-11-27
> > **Response to Reviewer fPix**
> >
> > Dear reviewer, this is a gentle reminder for our response above to answer your questions. We hope that we have addressed all of your questions, If there are any other questions or clarifications please let us know and we will answer those too asap. If you are satisfied with our responses then please consider updating your scores. Thanks again!

---

> > > ### Comment · Reviewer_fPix · 2022-12-10
> > > **Final recommendation**
> > >
> > > I thank the authors for their detailed response. Most of my concerns are addressed. However, other reviewers also point out several problems in the current version that affect the publication. Therefore, I keep my initial score.

---

### Official Review · Reviewer_7z7s · 2022-10-25

**Confidence:** 4
**Correctness:** 4
**Technical Novelty And Significance:** 3
**Empirical Novelty And Significance:** 3
**Recommendation:** 5

**Clarity, Quality, Novelty And Reproducibility:**

The whole paper is written clearly, and the work is with good reproducibility, especially considering the relationship with SupSup.
Although the relationship with SupSup may influence the significance of the novel, this work may still provide enough novel insights and observations.


**Strength And Weaknesses:**

Strength
- The work is well motivated. It starts from the limitation of the SupSup method and designs well-motivated and well-justified techniques to improve the supermask based method.
- The proposed method performs better than the compared methods with a similar setting.
- The paper conducted carefully designed analyses and discussions on the experiments.

Weakness
- The whole framework is based on SupSup with similar systematic limitations and benefits, which influences the significance at some level. Specifically, the task identifier is required in both training and testing, restricting the methods on the task-incremental setting.
- As a result, all the experiments are restricted to the task-incremental setting. Please correct me if I overlooked other settings.
- Some related works are not discussed, such as (but not limited to) the following ones. There are a series of approaches to generate subnetworks at a module or neuron level to alleviate the forgetting and parameter interference issue in CL. Although the settings may not be the same, they may be discussed.

    - Hurtado, Julio, Alain Raymond, and Alvaro Soto. "Optimizing reusable knowledge for continual learning via metalearning." Advances in Neural Information Processing Systems 34 (2021): 14150-14162.

    - Veniat, Tom, Ludovic Denoyer, and Marc'Aurelio Ranzato. "Efficient continual learning with modular networks and task-driven priors." arXiv preprint arXiv:2012.12631 (2020).

    - Yan, Qingsen, Dong Gong, Yuhang Liu, Anton van den Hengel, and Javen Qinfeng Shi. "Learning Bayesian Sparse Networks with Full Experience Replay for Continual Learning." In Proceedings of the IEEE/CVF Conference on Computer Vision and Pattern Recognition, pp. 109-118. 2022.
    - Golkar, Siavash, Michael Kagan, and Kyunghyun Cho. "Continual learning via neural pruning." arXiv preprint arXiv:1903.04476 (2019).

- How is the efficiency of the KKT module? It needs to run training and testing on multiple splits. Can the running time on the vision dataset be reported? How does the setting for the KKT module (like number of splits) influence the running time and performance?



**Summary Of The Paper:**

This paper proposes a mask-based approach to obtain subnetworks for specific tasks in continual learning (CL). It is an improvement of the previous method SupSup, which finds fixed supermask for each task to alleviate forgetting. This paper proposes to perform exclusive and nonoverlapping subnetwork training to avoid conflicting updating on the shared parameters. A specific KNN based method is proposed to enhance parameter sharing. The experiments show that the proposed method can produce better performance than other methods.

**Summary Of The Review:**

As discussed above, the work is well motivated and justified, and the paper is well-written.
The paper provides enough novel insights and observations and some empirical experimental results and analyses on the proposed method. The significance is influenced by its relationship with SupSup and that it is restricted to the task-incremental setting, where task identifiers are required in both training and testing.

---
I appreciate the authors' response, which addressed part of my concerns, especially on the KKT module. I am still concerned by the novelty (mainly about the relationship with SupSup) and technical contributions. After further discussions, I am further convinced that the experiment part also needs to be improved, especially in comparison with other parameter isolation-based methods, instead of only SupSup.
The work may be further improved by fixing the issues and conducting some more work, such as some attempts to extend the method to other settings beyond the task-incremental setting.

---

> ### Author Response · Authors · 2022-11-15
> **Response to Reviewer 7z7s**
>
> **Restriction to Task Incremental Setting:** Firstly, similar to SupSup, our idea could be extended to the case where the task identities are not provided during the inference. The SupSup paper presents a way to do this in their Section 3.3 and equation 4, where they propose to minimize entropy to select the best mask during inference. This can be directly applied to ExSSNeT in a setting where the task identities are not provided during inference. This lies outside the scope of our problem statement which was to improve the performance of supermasks for CL as also noted by you in the statement, “*The work is well motivated. It starts from the limitation of the SupSup method and designs well-motivated and well-justified techniques to improve the supermask-based method –Reviewer 7z7s*”
>
>
> **Value of Task Incremental Setting in NLP and Practical Applications:** Secondly, we would like to argue that this paradigm of known task identities is extremely popular in NLP because of the way these models are used in the current practical scenarios. There are significant challenges like knowledge sharing and scaling to many tasks in this case as well. Hence, we argue for the importance of this setting for practical use cases. Moreover, in Table 1, we compare with popular parameter isolation methods like prompt tuning and parameter-efficient adaptors in the setting of known task identities.
>
>
> Lastly, the value of our work lies in the simplicity of the idea that is applicable to both text and vision domains with strong results and through analysis of the method and its performance. Hence, we believe our work will be of value to the research community as well as practitioners.
>
>
> **Discussion with other related work:** Thanks for pointing out the works, we will discuss them in the final paper.
>
> **Efficiency, Runtime, and hyperparameters of the KKT module:** First, the KKT module is lightweight and efficient because it only runs once for each task before we start training on it and only uses a few batches to estimate the relevant mask. Given that we perform multiple epochs over the task’s data, the cost of the KKT module becomes negligible in comparison to it and runs in almost similar clock time as without it. The runtime on splitcifar100 datasets with 100 epochs for ExSSNeT is 168 minutes and for ExSSNeT + KKT is 173 minutes which is a very small difference.
>
>
> Second, as mentioned in Section 3.2, the number of learned KNN models is always the same as the number of learned tasks up till that point and is not a hyperparameter.
>
>
> Third, there are two main hyperparameters in the KKT module – (1) k for taking the majority vote of top-k neighbors, and (2) the total number of batches used from the current task in this learning and prediction process. We present additional results on the splitcifar100 dataset when changing these hyperparameters one at a time.
>
>
> In this experiment, we use 10 batches for KKT with a batch size of 64, resulting in 640 samples from the current task used for estimation. We report the performance of ExSSNeT when varying k. From this table, we observe that the performance increases with k and then starts to decrease but in general most values of k work well.
>
> | K | 1 | 5 | 10 | 20 | 50 |
> |---|---|---|---|---|---|
> | Test Accuracy | 71.38 | 71.66 | 71.01 | 70.46 | 69.74 |
>
>
>
> In this next experiment, we use a fixed k=10 and vary the number of batches used for KKT with a batch size of 64 and report the performance of ExSSNeT. We observe that as the number of batches used for finding the best mask increases the prediction accuracy increases because of better mask selection. Moreover, as few as 5-10 batches work reasonably well in terms of average accuracy.
>
> | B | 2 | 5 | 10 | 50 | 100 |
> |---|---|---|---|---|---|
> | Test Accuracy | 70.65 | 70.63 | 71.01 | 71.07 | 71.6 |
>
>
> From both of these experiments, we can observe that carefully selecting hyperparameters can lead to further improvement over the reported numbers in the paper but the KKT module is fairly robust to different values of these hyperparameters.
>
>
> We hope that we have addressed your main concerns and if so then please consider increasing your score. Thank you so much for your time and valuable comments!

---

> > ### Author Response · Authors · 2022-11-27
> > **Response to Reviewer 7z7s**
> >
> > Dear reviewer, this is a gentle reminder for our response above to answer your questions. We hope that we have addressed all of your questions, If there are any other questions or clarifications please let us know and we will answer those too asap. If you are satisfied with our responses then please consider updating your scores. Thanks again!

---

> > > ### Comment · Reviewer_7z7s · 2022-12-11
> > > **Final recommendation**
> > >
> > > I appreciate the authors' response, which addressed part of my concerns, especially on the KKT module. I am still concerned by the novelty (mainly about the relationship with SupSup) and technical contributions. I keep my initial score.

---

### Official Review · Reviewer_XDRc · 2022-10-28

**Confidence:** 3
**Clarity, Quality, Novelty And Reproducibility:** The writing quality is ok. I think th…
**Correctness:** 3
**Technical Novelty And Significance:** 2
**Empirical Novelty And Significance:** 3
**Recommendation:** 5

**Strength And Weaknesses:**

Strength:
1.	The framework has adopted and expanded previous works’ solution and achieved better performance than listed benchmarks.
2.	The idea is straightforward and easy to reproduce.

Weakness:
1.	The listed image benchmark is relatively simple, hard to justify the effectiveness of the proposed solution for dealing with big dataset.
2.	One of the most important steps in this pipeline is the supermask learning. It needs to be learned when adding new task. The author has listed the operations after 3 tasks in figure 1. What will be the parameters complexity after N tasks? Will that be a problem in the long run of continuous learning.


**Summary Of The Paper:**

In this work, the author focused on developing the solution for mitigating the catastrophic forgetting problem in the continuous learning problem. The proposed framework is mainly composed of two components, the first component is exclusive and non-overlapping subnetwork weight training, The second component is the KNN knowledge transfer module. The authors have evaluated the proposed pipeline on several benchmarks and demonstrate the improvement over listed benchmarks.

**Summary Of The Review:**

My rating is between the borderline and weakly reject since the evaluation data is relative small. Not sure if it can be generalized to more complicated task with big data.

---

> ### Author Response · Authors · 2022-11-15
> **Response to Reviewer XDRc**
>
> **Effectiveness of proposed solution on Big Datasets:** The datasets presented in the paper are challenging and standard benchmarks used for CL in the Vision and NLP domains. The vision datasets used in the paper are widely used in popular prior works [1,2] that are our baselines and are already cited in our paper. Whereas for the NLP domain, apart from the standard WebNLP datasets used by previous works [3,4], we went a step further to create a large and challenging benchmark from the popular glue tasks with approximately 1 million samples from challenging tasks (see Section 4.1 - Datasets). As shown in Table 1, our model results in significant improvements on this benchmark as well.
>
>
> **Results on ImageNet (1000 classes and 14 Million images):** Moreover, in order to concretely address your concerns, we further ran experiments on ImageNet datasets with 100 classes, and 14 million examples and report the results below.
>
> In this experiment, we take the ImageNet dataset with 1000 classes and divide it into 10 tasks where each task is a 100-way classification problem. We report the results for ExSSNeT andSupSup (strongest vision baseline) here and observe a strong improvement of 6.7%. We promise to add the other vision baselines in the final version.
>
> | Method | Average Accuracy (forgetting) |
> |:---:|:---:|
> | SupSup | 68.07 (0.0) |
> | ExSSNeT | 74.77 (0.0) |
>
>
>
> **Parameter Complexity after N Tasks:** As mentioned in section 3.1.1 of the paper, the parameters required on the GPU do not scale with the number of tasks and we just need to maintain the model weights and a single set of scores which can be swapped with the scores of a new task when learning it. Hence, ExSSNeT can even learn 100 tasks as shown in Q5 and Table 4. For each task, on disk, we need to store a learned binary mask that is sparse and can be stored efficiently  (see section 3.3.1). Moreover, the runtime of our method is similar to SupSup as shown in Appendix Table 10.
>
>
> We hope that our response and experiments on the ImageNet dataset address your major concerns and we request you to consider increasing your score, Thank you so much for your time!
>
>
> **References**
>
> [1] Dark Experience for General Continual Learning: a Strong, Simple Baseline. Buzzega et al, NeurIPS 2020
>
>
> [2] Supermasks in Superposition. Wortsman et al, NeurIPS 2020
>
>
> [3] Continual learning for text classification with information disentanglement based regularization, Huang et al, NAACL 2021
>
>
> [4] Lamol: Language modeling for lifelong language learning. Sun et al, ICLR 2019

---

> > ### Author Response · Authors · 2022-11-27
> > **Response to Reviewer XDRc**
> >
> > Dear reviewer, this is a gentle reminder for our response above to answer your questions. We hope that we have addressed all of your questions, If there are any other questions or clarifications please let us know and we will answer those too asap. If you are satisfied with our responses then please consider updating your scores. Thanks again!

---

### Decision · Program_Chairs · 2023-01-20

**Decision:**

Reject

**Justification For Why Not Higher Score:**

See Summary Of AC-reviewer Meeting

**Justification For Why Not Lower Score:**

N/A.

**Metareview: Summary, Strengths And Weaknesses:**

The paper builds on top of an existing continual learning algorithm dubbed as SupSup,  which uses a randomly initialized, fixed base network (model) and finds a supermask for each new task that selectively keeps or removes each weight to produce a subnetwork. The authors observe a limitation in that the performance of supermask is sub-optimal due to the fixed weights restriction, and hence proposed ExSSNeT (Exclusive Supermask SubNEtwork Training), which performs exclusive and non-overlapping subnetwork weight training. The authors also propose a KNN-based Knowledge Transfer (KKT) module that dynamically initializes a new task's mask based on previous tasks for improving knowledge transfer. The transfer is disabled if the nearest task is so far.

Strengths
--------------

- The method is well written (Reviewer jWtC), easy to reproduce, and achieved better performance than the listed benchmarks (Reviewer XDRc, Reviewer 7z7s, Reviewer jWtC).
- The work is well-motivated, starting from the limitation of SupSup method (Reviewer 7z7s,Reviewer fPix,  )
-The paper conducted carefully designed ablations(Reviewer jWtC),  analyses, and discussions on the experiments ( Reviewer fPix)

Weaknesses
---------------

-  The listed image benchmark is relatively simple, hard to justify the effectiveness of the proposed solution for dealing with big dataset (Reviewer XDRc)
- More analysis is needed to test the scalability of the method as both the number of tasks and the number of classes increase over time (Reviewer XDRc).
- The technical contributions of the paper are quite limited (Reviewer fPix, Reviewer jWtC). The framework is based on SupSup with similar systematic limitations and benefits, which influences the significance (Reviewer 7z7s).
- The approach is mainly explored in the incremental task setting rather than class incremental. This limits the evaluation of the method's ability to distinguish similar yet different classes that appeared in different tasks (Reviewer 7z7s, Reviewer jWtC,  AC).
- There are a series of approaches to generate subnetworks at a module or neuron level to alleviate the forgetting and parameter interference issue in CL. Although the settings may not be the same, they may be discussed and experimented against (see references provided by Reviewer 7z7s).
- The description of the paper can be polished (Reviewer fPix).
- Not well-established claim that ExSSNet can learn 100 tasks is mainly thanks to SupSup have achieved (Reviewer fPix), also Limited Experimental validation: what happens in the extreme case where there are no free weights. How is the performance of using a small model like LeNet for SplitMNIST and ResNet18 for SplitCIFAR100. Furthermore, how is the performance of using a larger model like Resnet101? Can larger models learn more than 100 tasks since they have more parameter spaces? (Reviewer fPix),


In conclusion, the reviewers recommended
5: marginally below the acceptance threshold
5: marginally below the acceptance threshold
6: marginally above the acceptance threshold
3: reject, not good enough

leaning overall not to accept the paper in its current version.


**Summary Of Ac-Reviewer Meeting:**

This paper was borderline as the ratings were 5,6,6,3 before the discussion. We had a meeting to discuss the paper. First, we discussed all strengths and paused if there was any point of disagreement. Then we did the same thing for weaknesses. Now that we are all aware of all viewpoints. Then each participating reviewer is asked if he/she changed her/her recommendation. After discussion, Reviewer 7z7s learned not to accept the paper. Here is the list of collective thoughts that concludes that the paper has indeed a merit but not yet ready for publication in its current version.

1) missing class incremental results makes it hard to judge the disciminativeness of the approach for identifying classes across tasks
2) comparison with different parameter isolation methods like HAT (AAAI,21), PackNet, UCB (https://openreview.net/forum?id=HklUCCVKDB). They have shown they outperform one of these methods, but more comparisons are needed.
3) There seem not to be any wrong claims in the paper. , but with the current experimentation and version, the paper may not be sufficiently significant as a contribution to ML knowledge. Hence, more technical contributions can be helpful.
4) Show that they can estimate the task id instead of arguing that it is possible
5) The paper needs to be placed better within parameter isolation methods (experimentally and in discussion)

The AC and reviewers encourage the authors to revise the paper based on the comments and submit the paper to a future venue.